# Contrasting behaviour under pressure reveals the reasons for pyramidalization in tris(amido) uranium(III) and tris(arylthiolate) uranium(III) molecules

Amy N. Price [1,2], Victoria Berryman [3], Tatsumi Ochiai[1], Jacob J. Shephard[1], Simon Parsons [1✉],
Nikolas Kaltsoyannis [3✉] & Polly L. Arnold [1,2✉]

A range of reasons has been suggested for why many low-coordinate complexes across the periodic table exhibit a geometry that is bent, rather a higher symmetry that would best separate the ligands. The dominating reason or reasons are still debated. Here we show that two pyramidal $UX_3$ molecules, in which X is a bulky anionic ligand, show opposite behaviour upon pressurisation in the solid state. $UN''_3$ (**UN3**, $N'' = N(SiMe_3)_2$) increases in pyramidalization between ambient pressure and 4.08 GPa, while $U(SAr)_3$ (**US3**, SAr = S-$C_6H_2$-$^tBu_3$−2,4,6) undergoes pressure-induced planarization. This capacity for planarization enables the use of X-ray structural and computational analyses to explore the four hypotheses normally put forward for this pyramidalization. The pyramidality of **UN3**, which increases with pressure, is favoured by increased dipole and reduction in molecular volume, the two factors outweighing the slight increase in metal-ligand agostic interactions that would be formed if it was planar. The ambient pressure pyramidal geometry of **US3** is favoured by the induced dipole moment and agostic bond formation but these are weaker drivers than in **UN3**; the pressure-induced planarization of **US3** is promoted by the lower molecular volume of **US3** when it is planar compared to when it is pyramidal.

[1] EaStCHEM School of Chemistry and The Centre for Science at Extreme Conditions, The University of Edinburgh, King's Buildings, Edinburgh EH9 3FJ, UK. [2] University of California, Berkeley and Lawrence Berkeley National Laboratory, Berkeley, California, CA 94720, US. [3] Department of Chemistry, The University of Manchester, Oxford Road, Manchester M13 9PL, UK. ✉email: Simon.Parsons@ed.ac.uk; nikolas.kaltsoyannis@manchester.ac.uk; pla@berkeley.edu

Three-coordinate lanthanide and actinide complexes of the type $MX_3$ are typically pyramidal rather than the trigonal planar shape which would be anticipated to minimise steric congestion in complexes where bonding is primarily ionic[1–7]. This unexpected deviation from the highest symmetry is also widespread in low-coordination number p- and d-block complexes[8].

Electronic structure manipulation through geometry for f-block complexes is an emergent field, with potential applications in molecular- and quantum-scale sensing and computing applications[9]. For example, planarization of the three-coordinate lanthanide complex $K[Lu^{II}(OC_6H_2(Ad)_2-2,4-^tBu-6)_3]$ places the effective S = 1/2 spin in an s/d hybrid orbital that affords the molecule with one of the largest hyperfine interactions for a molecular system, $A_{iso} = 3,467 \pm 50$ MHz, and a very large associated clock transition, that make it competitive with a trapped ion qubit for quantum information storage[10].

Potentially competing factors have been suggested to account for an unexpected deviation from the geometry which should minimise steric overlap, such as pyramidalization, Fig. 1. The tendency towards pyramidalization can arise from the generation of a dipole moment that gives additional electrostatic stabilisation (the so-called polarisable ion model) (Fig. 1b), the optimisation of dispersion forces between peripheral groups on the ligands (Fig. 1c) or the formation of additional weak or agostic bonds between metal and ligand (Fig. 1d). Dispersion is increasingly recognised as providing a significant stabilising force in many unusual new p- and d-block molecules that have been reported over the past decade[11], and its effect increases rapidly (as the 6th power of distance) at short inter atomic separations. In a pyramidal geometry, the metal's d orbitals can only be involved in σ-type bonding on symmetry grounds, Fig. 1e, whereas a planar geometry enables involvement of metal d-orbitals in π-type bonding. This has been investigated computationally for a series of small X (H, Me, Hal, $NH_2$) in group 3 and 4 complexes[12]. However, for An which exhibit greater bonding covalency than group 3, and for heavier heteroatom-bound ligands, with more available X(p) π-orbitals, greater planarity would be expected from this argument. For example, the S-donor $MX_3$ molecules (X = SAr = $S-^tBu_3-2,4,6-C_6H_2$, M = U, La, Ce, Pr, Nd[13]; M = Sm[14]) are essentially isostructural with shallower pyramidal geometries, leaving the questions still unanswered.

The importance and effects of covalency in bonding in the actinides, as well as its definition[15], is a contemporary problem and has widespread ramifications in energy technologies such as f-block ion separations. For example, understanding the differences in M-X bonding with sulfur-based X ligands might help answer the fundamental question of why the commercial S-donor extractant chelator Cyanex301, bis(2,4,4-trimethylpentyl)dithiophosphinic acid, shows a separation factor of over 1000 for $Am^{III}$ over $Eu^{III}$ which have the same ionic radius and electronegativity[16,17].

Systems in which energetically competitive structures are related by subtle changes such as weak metal-ligand interactions, or ligand-ligand interactions may be sensitive to external conditions such as temperature or pressure. We studied the effect of high pressure (up to 5 GPa) on low-valent uranium complexes with bulky amido ($N'' = N(SiMe_3)_2$) and aryloxide (Odtbp) ligands[18]. We found, supported by Quantum Theory of Atoms in Molecules (QTAIM) and Natural Bond Orbital (NBO) calculations, evidence of genuine agostic U···H bond formation at high pressure.

In this work, we have undertaken variable pressure single crystal X-ray diffraction studies on two contrasting pyramidal compounds, Fig. 1; the highly symmetrical, uranium(III) tris(amide) $U(N(SiMe_3)_2)_3$ (**UN3**) and the less well-studied, more polarisable $U(SAr)_3$ (SAr = arylthiolate $S-^tBu_3-2,4,6-C_6H_2$) (**US3**)[13,19]. These are combined with NBO and QTAIM analyses of DFT-derived electronic structures, in order to evaluate the structural drivers set out in Fig. 1[20,21]. The study of uranium rather than rare earth $MX_3$ complexes enables us to explore all the potentially contributing factors hypothesised to favour pyramidalization. The actinide can make use of four different types of atomic orbitals (s, p, d and f) in metal-ligand bonding, in contrast to lanthanides, where f-orbital participation in bonding is tiny[22].

## Results and discussion

**Single crystal X-ray diffraction studies of US3 and UN3.** Complexes **UN3**[23,24] and **US3**[13] were prepared and the single crystal X-ray structures of **UN3** as its cyclohexane solvate and **US3** were determined at ambient temperature and pressure, and then at increased pressure steps up to 4.09 GPa and 5.25 GPa, respectively. Figure 2 summarises the changes observed.

The experimental structure of **UN3** shows uranium disordered equally over two positions above and below the plane of the three heteroatoms, Fig. 2. At ambient pressure six U-C(H) metal-ligand close contacts in the cavity above the pyramid (U-C range 2.943(7)–3.301(8) Å) are found. With increasing pressure, pyramidalization increases, evidenced by the $U-N_3$ oop distance increasing by 0.075(3) Å. There is also a significant shortening of the U-C(H) close contacts. In the solid state, **UN3** has unequal U-N-Si angles of 110.17(10)° and 123.60(11)° since the Si-C bond of one of each amido ligands is closer to the U centre [U···$C_\gamma$ 3.055(5) Å; U···Si 3.2800(12) Å]. At 4.09 GPa the closest U-C(H) contact distance of 3.055(5) Å (there are three, still equivalent by symmetry) shortens to 2.939(7) Å (Fig. 3), and the corresponding U···Si shortens to 3.2809(8) Å. These can be attributed to stabilisation by agostic M···Si-$C_\gamma$ interactions, similar to $[U\{CH(SiMe_3)_2\}_3]$ and SmN3[4,25]. The U-N bond length does

### a) Out of plane (oop) distance is used as a measure of pyramidalization

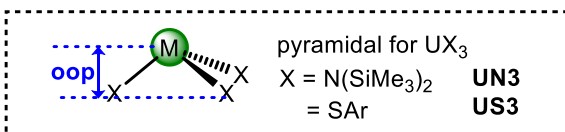

pyramidal for $UX_3$
X = $N(SiMe_3)_2$  **UN3**
= SAr  **US3**

b) molecule's dipole moment

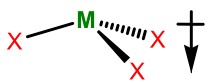

c) X-X weak interactions / dispersion forces

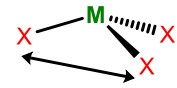

d) M-X weak / agostic interactions

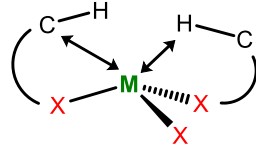

e) lower M(d)-X(p) π overlap

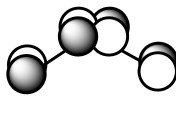

**Fig. 1 Summary of the pyramidal molecules studied in this work and the factors thought to contribute to pyramidalization in Ln and AnX3 complexes. a** Out of plane (oop) distance between the M and the X3 plane is described as oop, shown in blue, is used to provide a simple measure of pyramidalization in US3 and UX3. Contributing factors to pyramidalization in MX3 include **b** dipole moment, **c** ligand-ligand interactions, **d** metal-ligand agostic interactions and **e** M(d)-X(p) π overlap. UN3 = $U(N(SiMe_3)_2)_3$ and US3 = $U(SAr)_3$ where SAr = arylthiolate $S-^tBu_3-2,4,6-C_6H_2$.

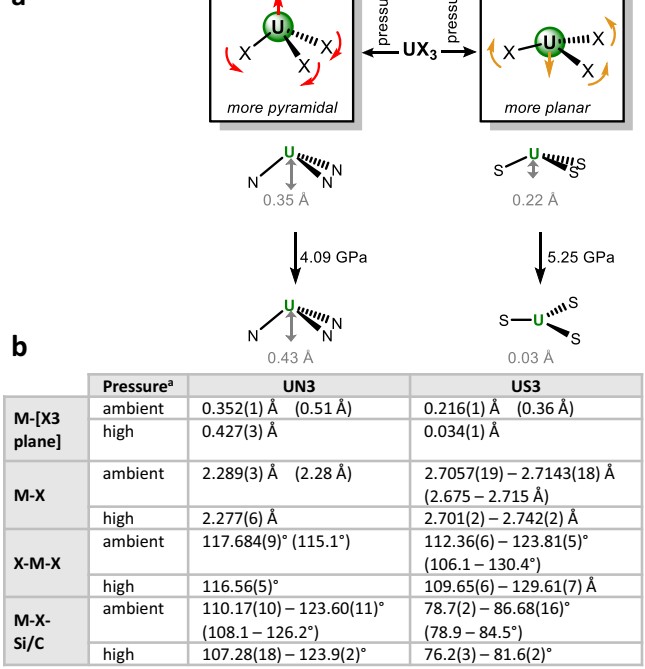

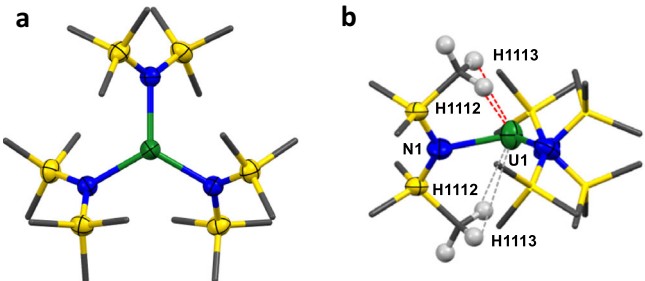

a. high = 4.09 GPa for **UN3**, 5.25 GPa for **US3**

**Fig. 2 The change in pyramidalization observed for UN3 and US3 with pressure. a** Increased pyramidalization of **UN3** with pressure, and planarization of **US3** with pressure. **b** Key distances and angles and their change with pressure for the two compounds, with computed values in brackets. **UN3** = $U(N(SiMe_3)_2)_3$ and **US3** = $U(SAr)_3$ where SAr = arylthiolate S-$^tBu_3$–2,4,6-$C_6H_2$.

a | b

**Fig. 3 High pressure structure of UN3. a** Solid state structure of UN3 from top at 4.09 GPa, **b** side view of the solid-state structure of UN3 showing pyramidalization and U-CH agostic interactions at 4.09 GPa (red = close contacts, grey = longer contacts). UN3 = $U(N(SiMe_3)_2)_3$. Atom colour code: Green = U; blue = N; gold = Si; grey = C.

not change significantly with pressure (from 2.289(3) Å at ambient pressure to 2.277(6) Å at 4.09 GPa).

At ambient temperature and pressure, Fig. 4, **US3** is essentially isomorphous with the 100 K structure reported for it and Ln = La, Ce, Pr, Nd, Sm[13,14]. The U-S₃ oop distance is 0.216(1) Å with a propeller-like SAr arrangement. In the ambient pressure structures of **US3** and all the **LnS3**[13], except **SmS3** which contains a smaller metal[14], there is a small boat-distortion of the arene ring, with the *ipso*-carbon the furthest from the ring plane (0.070 to 0.086 Å).

The U-S distances in **US3** (range 2.7057(19)–2.7143(18) Å) are similar to those in Meyer's recently reported U^III adduct of a chelating tris-thiophenolate (U-S distance = 2.7083(8) Å), both of which are much shorter than the few other U^III-S compounds in the literature (range 2.77–2.87 Å)[19,26–28]. The M-S-C_ipso angles in

**US3** are more acute (78.7(2)–86.68(16)°) than all including the tris-thiophenolate (92.31(9)°)[27,28]. The short U-C contacts in **US3** are benzylic-type interactions with the *ipso* and one *ortho* carbon in each ligand ring (Fig. 3, top-left) (average U-C_ipso 3.045(115) Å, average U-C_closest ortho 3.279(89) Å at ambient T, p). The alternating ring bond lengths further support this, and the effect is more prominent for U than for the Ln congeners, in agreement with the use of more diffuse orbitals by uranium.

**US3** also has several short U-H contacts, (range 2.47 to 2.82 Å), which were characterised as agostic interactions through NBO analysis by Ephritikhine and co-workers (see below). In contrast to **UN3**, these favourable U-HC agostic interactions can form both above and below the **US3** core. The crystallography suggests that there are four short S-HC contacts within each ligand (range 2.55–2.67 Å), (Fig. 3, top-right). Weak sulfur-containing hydrogen bonds (SCHBs) can play a cumulative stabilising and structure-directing role[29–33], but our calculations do not find that they play a noticeable role in this case (see below).

Planarization with increasing pressure is evidenced by the U-S₃ oop distance decreasing from 0.216(1) Å to 0.034(1) Å at 5.25 GPa, Fig. 3, bottom. The average U-S distance does not significantly change (2.7109(5) at 0 GPa to 2.717(22) Å at 5.25 GPa) although the difference between shortest and longest increases from 0.009 Å to 0.041 Å. Instead, the U-S-C_ipso angle becomes more acute; from a range of 78.7(2)-86.68(16)° (ambient) to 76.2(3)–81.6(2)° (5.25 GPa). The average U-C_ipso distance shortens from 3.045(115) Å to 2.950(82) Å, and the average U-C_ortho distance shortens even more dramatically from 3.279(89) Å to 3.048(161) Å by 5.25 GPa. Some of the S…HC contacts shorten to <2.4 Å as the $^tBu$ groups rotate to improve packing; these are well within the sum of the van der Waals bond radii of sulfur and hydrogen (3.0 Å).

**Computational studies.** Computational studies were carried out on structures derived from the crystallographic data. The geometries of **UN3** and **US3m**, where **US3m** is a computationally more tractable model for **US3** in which the para $^tBu$ groups are replaced by H atoms, were calculated using scalar relativistic, hybrid density functional theory (PBE0), incorporating Grimme's dispersion corrections with the Becke-Johnson damping parameters. Both systems optimise to pyramidal structures, confirmed as true minima by harmonic vibrational frequency analysis. Geometry optimisation of **US3** and **US3m** using a smaller basis set confirmed that the presence of the para $^tBu$ groups has minimal structural effect: pyramidalization varied by 0.01 Å and U-X distances by no more than 0.008 Å. Further calculations located planar at U (**UN3**) or near planar at U (**US3m**) transition state structures, each possessing a single imaginary vibrational mode associated with interconverting two pyramidal conformers, with modest energy barriers of 5.6 and 9.8 kJ mol⁻¹ respectively, confirming that a planar geometry is achievable. For comparison, the experimentally determined inversion energy for NH₃ is 24.2 kJ mol⁻¹ [34].

The computed dipole moments at the optimised pyramidal geometries are large for both **UN3** and **US3m**, 2.54 and 2.03 D, respectively. For comparison, experimentally determined values in bent p-block molecules are rather lower (H₂O: 1.84, NH₃: 1.44, PH₃: 0.55, PCl₃: 1.16 D)[35]. At the transition states, the dipole moments are much reduced, to 0.43 and 0.60 D respectively for **UN3** and **US3m**. The large dipole moments for the pyramidal forms of both systems will contribute to the stability of those geometries, and the reduction at the transition states will oppose planarization, albeit to a significantly lesser extent in **US3m** than **UN3**.

In their 2006 contribution, Roger et al. used second order perturbation theory analysis within the NBO framework to explore

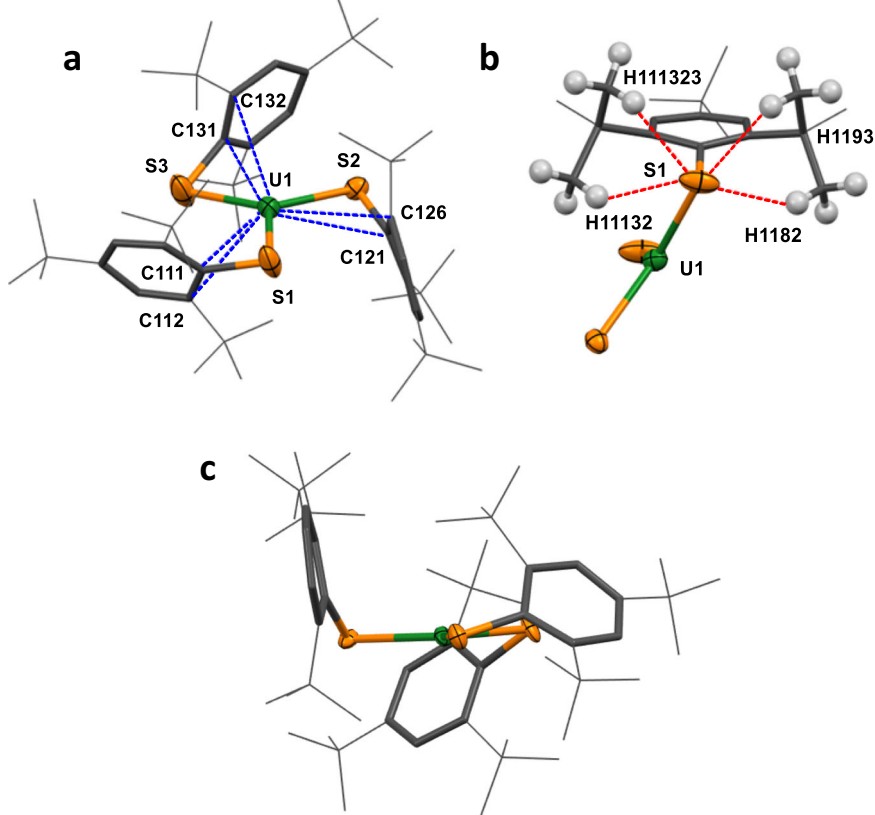

**Fig. 4 Solid state structure of US3 under ambient conditions and at high pressure. a** Solid state structure of **US3** at ambient pressure with intramolecular U-C contacts drawn in blue, **b** solid state structure of **US3** at ambient pressure with intramolecular S-H contacts drawn in red, **c** solid state structure of **US3** at 5.25 GPa along the US₃ plane. Atom colour code: Green = U; orange = S; grey = C; white = H. **US3** = U(SAr)₃ where SAr = arylthiolate S-$^t$Bu₃−2,4, 6-C₆H₂.

the electronic structure of a model for **US3**. They concluded that there are stabilising interactions resulting from donations from (i) a $C_{ipso}$-$C_{ortho}$ bond to empty U f (ii) a S-$C_{ipso}$ bond to empty U f and (iii) $^t$Bu C-H to empty U d/f (an agostic effect enhanced by slight hyperconjugation between $C_{ipso}$-$C_{ortho}$ and C-C of $^t$Bu). They also identified delocalisation from a S lone pair to $C_{ipso}$-$C_{ortho}$ π* bond. Unfortunately, only the agostic effect was quantified, providing "a stabilisation of roughly c. 5 kcal mol⁻¹". We have conducted a similar analysis on both the pyramidal and transition state structures of **UN3** and **US3m**, details of which are collected in the SI. Overall, we find quite different results for **US3m** that those of Roger et al., most likely arising from substantial advances in computational methodology over the last 15 years. The principal difference between pyramidal and transition state **US3m** is a significant reduction in agostic interactions in the latter, indicating that the high pressure behaviour of **US3** is not the result of enhanced agostics. Note that there are no sulfur containing hydrogen bond interactions in either form of **US3m** larger than the NBO default cut off of 0.25 kcal mol⁻¹ (1.046 kJ.mol⁻¹). Although QTAIM analysis does locate S-H bond paths and bond critical points, the electron and energy densities at these critical points are very low (< 0.02 and c. 0.000 au respectively) suggesting that these interactions are at most very weak, in agreement with the NBO perturbation theory analysis. In **UN3**, the energies associated with the NBO donor-acceptor interactions are much less than for the sulfur system, though we do find evidence of slightly stronger agostic interactions in the transition state than the pyramidal form. As with **US3m**, these agostics do not appear to determine the high-pressure behaviour of **UN3**. The boat-distortion in the ligand arenes

mentioned above for **US3**, which was previously ascribed to the benzylic U-C interactions disrupting the bonding in the ring, is attributed in these calculations to donation from the U-S bonds to the $C_{ipso}$-$C_{ortho}$ π* orbital of the arene.

To assess the U-N and U-S bonding, and in particular the way in which it changes between the pyramidal and transition state structures, we have analysed both Quantum Theory of Atoms in Molecules metrics and Natural Localised Molecular Orbital composition data. These are presented and discussed in the SI, page 18. While there are clear differences in the bonding between **UN3** and **US3m**, there are only very minor changes between the pyramidal and transition state structures (with marginally weaker bonding in the latter), from which we conclude that bonding changes (including in the d-orbital contribution to the NLMOs, and the metal-ligand overlap) are insufficient to drive the differences in the high pressure behaviour of **UN3** and **US3m**.

**Volume discussion.** The molecular volumes (isodensity surface of 0.001 electrons/Bohr³) of **UN3** and **US3m** were determined for the optimised pyramidal and transition state structures. For **UN3**, there is a small increase, 1% between the pyramidal and planar forms. Experimentally, the volume defined by the van der Waals surface of the pyramidal molecules of **UN3** in the experimental crystal structures shows a modest reduction between ambient pressure and 4.80 GPa (Fig. 15 in the SI), but a systematic trend is rather difficult to discern, with most points scattered within 0.5% of the ambient pressure volume. The combination of the computational and experimental data implies that the molecular volume of the pyramidal form of **UN3** is quite insensitive to

pressure, but would increase slightly on becoming planar. A planar form of **UN3** would therefore not be promoted by application of pressure.

By contrast, the molecular volume of **US3** in the experimental crystal structures shows a linear decrease between ambient pressure and 5.25 GPa, with a 2% reduction in volume as the molecule changes from a pyramidal to a planar geometry (Fig. 4). The volume reduction in the theoretical structures is still more marked, with the volume of the 0.001 electrons/Bohr$^3$ isosurface of **US3m** being 7% smaller in the planar than in the pyramidal form. A volume reduction of this magnitude would strongly stabilise the planar form relative to the pyramidal alternative at high pressure.

*Summary.* To summarise, upon pressurisation in the solid state, **UN3** becomes more pyramidal, whereas **US3** becomes planar. Until now, it has been presumed that larger ligands are needed to enforce a planar geometry[36], but sterics are not the dominant factor in determining the geometry of these complexes. The SAr ligand is significantly smaller than the N(SiMe$_3$)$_2$ ligand, and at least for the rare earth elements and the actinides, multiple factors beyond sterics need to be considered when choosing ligands that can usefully control geometry in unusually low-coordinate complexes. The opposing pressure responses of **UN3** and **US3** have helped to reveal the dominant reasons why the molecules' normally favoured conformation is pyramidal. The increasing pyramidalization of **UN3** with pressure is favoured by an increased dipole moment and reduction in molecular volume, while the planarization of **US3** with pressure is promoted by the slight reduction in molecular volume achievable. We find that agostics do not determine the pressure behaviour for either molecule and that the bonding changes that occur with changes in the degree of pyramidalization of **UN3** or **US3** (including in the d-orbital contribution to the NLMOs, and the metal-ligand overlap) are insufficient to drive the differences in their high-pressure behaviour.

## Methods

UN$_3$ and US$_3$ were synthesised according to literature procedures: further details are included in the supplementary information file under the methods section[1,2]. Diffraction data at ambient pressure were collected at 270 K with the crystal mounted under a film of oil protect the crystal from the air while maintaining a steady position on its mount. Data sets at high pressure were collected in a Merrill-Bassett diamond anvil cell using Boehler-Almax cut diamonds, a tungsten gasket and ruby as a pressure marker. The solubility of the compounds necessitated the use of fluorinert (FC70) as the medium between 0 and 1 GPa. Fluorinert has a hydrostatic limit of 0.6 GPa[36], and so no data could be collected between 1 and 2.37 GPa. Above 2 GPa, pressure was applied using a 1:1 mixture of pentane and isopentane as a hydrostatic medium. The maximum pressure reached was 4.09 Gpa for **UN3** and 5.25 Gpa for **US3**. Data at ambient pressure and 0.87 GPa were collected using MoKα radiation, those at higher pressures were collected at Diamond Light Source on Beamline I19 (Experimental Hutch 2) with X-rays of wavelength 0.4859 Å. Shaded regions of the detector were omitted during integration of the diffraction images, and a multi-scan correction was used to account for absorption, gasket shading and other systematic errors.

## Data availability

The crystallographic data generated in this study have been deposited in the CCDC database under deposition numbers 2143670-2143686. Copies of the data can be obtained free of charge via https://www.ccdc.cam.ac.uk/structures/. The calculated atomic coordinates for **UN3** and **US3m** generated in this study are provided in the Supplementary Information. Further information regarding the synthesis of **UN3** and **US3**, as well as further analysis of their crystal structures under both ambient pressure and higher pressure is provided in the Supplementary Information.

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

## Acknowledgements
We thank the University of Edinburgh and the EPSRC for funding through grants EP/N022122/1 and EP/N021932/1 (A.N.P., V.B., T.O., J.J.S., S.P., N.K., P.L.A.). This project has also received funding from the European Research Council (ERC) under the European Union's Horizon 2020 research and innovation programme (grant agreement No 740311; PLA). We thank Dr Mark Warren for his assistance during synchrotron beam time, the Diamond Light Source and STFC for provision of synchrotron beam-time (MT16139), and the University of Manchester's Computational Shared Facility for access to computing resources and associated support services. We thank the Japan Society for the Promotion of Science for International Fellowship funding to TO. Additional discussion, analysis, and writing of this manuscript (A.N.P., T.O., P.L.A.) was supported by the U.S. Department of Energy (DOE), Office of Science, Office of Basic Energy Sciences, Chemical Sciences, Geosciences, and Biosciences Division at the Lawrence Berkeley National Laboratory under Contract DE-AC02-05CH11231.

## Author contributions
A.N.P. analysed data and wrote the SI. V.J. performed calculations, T.O. prepared **US3** and **UN3**, J.J.S. mounted the samples, collected the X-ray data and analysed the crystal structures. S.P., N.K. and P.L.A. designed the study. All authors contributed to discussions about the data and co-wrote the manuscript.

## Competing interests
The authors declare no competing interests.
