## [Peer Review File · Nature Communications]

REVIEWER COMMENTS

Reviewer #1 (Remarks to the Author):

I have reviewed "Contrasting behaviour under pressure reveals the reasons for pyramidalization in tris(amido)uranium(III) and tris(arythiolate) uranium(III) molecules" by Price and co-workers and am pleased to recommend it for publication. The paper describes an interesting problem - the geometric preferences in actinide complexes bearing multiply bonded ligands - and the authors bring to bear a powerful suite of tools combining synthesis, high pressure crystallography and computational methods to explain the observed structures. Naively, I would have argued that metal-to-ligand π bonding was the critical element of these structures (I would be wrong) and, to my delight, the authors demonstrate convincingly that a combination of polarization, volume minimization and agostic interactions are the key drivers of the observed structures. It's refreshing to see the precision obtainable, for example, of the pyramidal inversion and detailed breakdown that the contributions from agostic interactions make (it was not possible/practical not that long ago).

With these plusses out of the way, here are a few points the authors may wish to consider:

Scheme 1 (MS, p.4), shows a reduction in the M-N bond distance and a lengthening of the M-S bond length with increased pyramidalization. Are these both interpreted as a changes of M(d)-X(p) π overlap as suggested in Figure 1(d)? It looks like the M-N π bonds are getting stronger (shorter, higher % contrib. from metal d orbitals) and, regardless, the π system is more complicated than this single figure suggests. True, the in-plane overlap changes with increased pyramidalization, but so does the nature of the d orbitals involved in the bonding. When you depart from D_{3h}, the two sets of doubly-degenerate d-orbitals merge into a set of four, and all can participate in π bonding.

On S-17, the authors describe their NBO analysis. I agree with the outcome but the assignment of 1 σ and 1 π bond per amide group is valid only for the pyramidal structure and not the trigonal planar one. In D_{3h} symmetry, the in-plane π orbitals transform as $a_2' + e'$ (see JACS 1990, 112, 1642 for example) and, since it is the d-orbitals that are more important for bonding in early actinides, there are only 2 π

bonds that are possible (of e' symmetry). It would not be problematic if f orbitals were used as they span both representations, again affirming the importance of d-orbitals in early actinides. That the Wiberg bond index, delocalization index and electron density at the bond critical point show increased bonding in the pyramidal UN₃ and US₃ structures is consistent with these n effects. Was there really zero participation of f orbitals in the pyramidal and planar structures?

Bottom line: great paper.

Mark Schofield

Reviewer #2 (Remarks to the Author):

The solid-state structures of two U(III) complexes, [U{N(SiMe₃)₂}₃] and [U{S-2,4,6-tBu₃C₆H₂}₃], were examined under high pressure from 0 GPa to 4.09 GPa (for the amide) or 5.26 GPa (for the thiolate). The tris(amide) shows more pyramidalization while the tris(thiolate), surprisingly, shows more planarization under high pressure. Computational methods show this is not due to agostic interactions, as previously shown in the tris(thiolate) complex, but pyramidalization is due to increased dipole and reduction in the volume of the molecule. While planarization in the tris(thiolate) is due to reduction in molecular volume. As the authors points out, planarity is commonly thought to be achieved through more sterically demanding ligands, but this is not the case here with the amide being larger than the thiolate. Overall, I only have some minor changes, and support its publication as it reminds the f element community that things are not just sterically driven.

Minor issues:

1. Caption of Figure 1 - For clarity, I suggest that the authors change to 'out of plane (oop) distance between the M and and the X₃ plane, shown in blue,....' I read oop and thought it was a mistake as, at my age, it is usually something I say after sneezing, but that is another matter. I do not believe that oop is defined in the text and probably should be as well.

2. On page 5, SMes* is used as the abbreviation which is correct, just define this earlier for a more general audience who may not know the tris(tert-butyl) phenyl is Mes*.

3. The major thing for the authors to chew on is do the authors think that these interactions are agostic or anagostic as defined by Brookhart, Green, and Parkin (PNAS 2007, 104, 6908)? The M-H distances and angles are for transition metals so those distances would naturally be longer with an f element metal center, but the definition of agostic interaction of having some covalent character in a M-H, I would argue, would not apply to any f element compound except with a hydride ligand. Maybe the authors have stronger opinions and it makes no difference if they do not wish to change since it is ingrained in the literature. Again, something for the authors to chew on.

Reviewer #3 (Remarks to the Author):

The authors have presented a thorough and very detailed study regarding U(III) complexes with N- and S-donors. They have used SC-XRD and quantum chemical calculations to analyse the complexes' structural and electronic alterations when applying pressure. They could separate and quantify possible factors influencing a pyramidal vs. planar geometry of the UX₃ compounds, presenting a unique example of a U(III) S-donor complex showing a tendency of planarization upon pressurisation. The provided manuscript comprises a variety of reliable and very interesting data using non-standard equipment and addressing highly relevant questions in actinide research. The reviewer had high expectations of the outcome and of the kind of conclusions that might be drawn from the studies, particularly as the introduction also suggested to give answers to urgent questions, e.g. to actinide covalency. However, the expectations could not fully be met. For example, it is noted in the introduction that immense differences in covalency had already been observed when comparing Eu(III) with Am(III). This raises the question why the authors haven't included e.g. Nd(III) complexes NdN₃/ NdS₃ as lanthanide analogues for U(III) in order to get one step closer to resolving this issue (or La(III) for even more comparable ionic radii). Similarly, the reviewer has asked her-/himself why UN₃ was compared with US₃, not N- vs. P-donors or O vs. S. Thus, we reviewer would wish to find a slightly revised introduction explaining the necessity of studying geometric changes in An compounds bearing in mind the outcome of this manuscript. In other

words, seeking an answer to a clear question at the beginning would help to keep an inner logic of the whole story.

Still, it is a fascinating and highly relevant work and opens up a broad range of possibilities for future research.

My detailed comments including some typos and minor wording issues to the manuscript are the following:

- 1) In the author's line, a comma is missing after Nikolas Kaltsoyannis.
- 2) Abstract, line 5: There seems to be an unnecessary comma after (UN₃, N" = N(Si(me₃)₂), and after "ambient" there should be "pressure" included.
- 3) Page 2, line 3: ..."to account for this..." "this" should be replaced by e.g. "an" as reference is lacking.
- 4) Introduction, line 14: It should be X(p) pi (with the pi outside the bracket)
- 5) Introduction, line 15, bracket: M = S ... Should it be S_m instead?
- 6) In the introduction, factors for increasing the tendency for pyramidalization are listed. What about strong metal-ligand bonds?
- 7) Page 3: The single images and the respective text in Figure 1 should be aligned.
- 8) Page 4, Scheme 1: Here, a table with i.a. bond length ranges for UN₃ and US₃ are given. Please double-check the upper value of M-X for US₃ at ambient pressure (2.7143(18)), in comparison with Table S4 (here, it is 3.035(6) Å).
- 9) Sometimes, there is a space between value and Å, sometimes not. Please unify.
- 10) Figure 3: Something went wrong in the formatting, I guess... Single images should be sorted and aligned.
- 11) Page 6
line 5: tris-thiophenolate instead of trithiophenolate
line 6: It most probably should be "M-S-Cipso" angles? (in the SI, page S-9, it is also displayed like this)
lines 8/9: Reference error for Fig 3
line 9: Missing space in ", top-left)(average..."
line 9: The pressure symbol is a small p (...at ambient T, P)
line 24: The "t" in tBu should be superscript in the whole text.
line 27 et seq.: It should be mentioned that many calculations were carried out, for ambient but also for high pressure structures.
- 12) It has puzzled me, that US₃ was reduced in size for computational studies to US_{3m}. Of course, fewer atoms mean significant reduction of computational costs. On the other hand, US₃ is not super big, removal of three electron donating groups will

have an electronic impact, and, most importantly, potential agostic interactions would form between the C-H groups of the tBu groups and U, as described in the article. In order to study if agostic interactions play a role or not, tBu groups cannot be removed.

Thus it is not surprising that significantly reduced agostic interactions were found in US3m vs UN3 (page 7, line 20 et seq.). In addition, it is no surprise that sulphur containing hydrogen bond interactions also don't play a role in US3m, as they would come from tBu H's, too (page 7, line 25 et seq.).

13) Page 7, line 19: The "pi" symbol is not displayed.

14) Page 8, line 12 et seq.: What is the driving force for volume reduction? At least in my understanding it is rather a response to an electronic or magnetic effect. Could you relate the reduced volume e.g. to reduced dipole or electrostatic interactions? Maybe ELF plots can give an answer here in order to visualize electron density changes between ambient and high pressure structures?

15) Page 9, lines 5&6: The introduction suggested an impact of bonding changes and planarization...

16) Supporting information: A lot of additional data and information is given here, what is really nice and supportive. Please double-check formatting, also including figures and tables.

Page S-7, 2nd paragraph, 2nd sentence: "At ambient pressure...". Here, "The structure at ambient pressure" would make more sense.

Reviewer response document

Reviewer #1 (Remarks to the Author):

I have reviewed "Contrasting behaviour under pressure reveals the reasons for pyramidalization in tris(amido)uranium(III) and tris(arylthiolate) uranium(III) molecules" by Price and co-workers and am pleased to recommend it for publication. The paper describes an interesting problem - the geometric preferences in actinide complexes bearing multiply bonded ligands - and the authors bring to bear a powerful suite of tools combining synthesis, high pressure crystallography and computational methods to explain the observed structures. Naively, I would have argued that metal-to-ligand π bonding was the critical element of these structures (I would be wrong) and, to my delight, the authors demonstrate convincingly that a combination of polarization, volume minimization and agostic interactions are the key drivers of the observed structures. It's refreshing to see the precision obtainable, for example, of the pyramidal inversion and detailed breakdown that the contributions from agostic interactions make (it was not possible/practical not that long ago).

With these plusses out of the way, here are a few points the authors may wish to consider:

1. Scheme 1 (MS, p.4), shows a reduction in the M-N bond distance and a lengthening of the M-S bond length with increased pyramidalization. Are these both interpreted as a changes of M(d)-X(p) π overlap as suggested in Figure 1(d)? It looks like the M-N π bonds are getting stronger (shorter, higher % contrib. from metal d orbitals) and, regardless, the π system is more complicated than this single figure suggests. True, the in-plane overlap changes with increased pyramidalization, but so does the nature of the d orbitals involved in the bonding. When you depart from D3h, the two sets of doubly-degenerate d-orbitals merge into a set of four, and all can participate in π bonding.

response: The M-N distance does shorten, but by only 0.01 Ang, which is a very small amount. The U-S bond length changes differ – U-S1 lengthens by circa 0.03 Ang., while the other two shorten by less than 0.01 A. Thus it's hard to see how this is explained by π -overlap arguments.

2. On S-17, the authors describe their NBO analysis. I agree with the outcome but the assignment of 1 σ and 1 π bond per amide group is valid only for the pyramidal structure and not the trigonal planar one. In D3h symmetry, the in-plane π orbitals transform as $a_2' + e'$ (see JACS 1990, 112, 1642 for example) and, since it is the d-orbitals that are more important for bonding in early actinides, there are only 2 π bonds that are possible (of e' symmetry). It would not be problematic if f orbitals were used as they span both representations, again affirming the importance of d-orbitals in early actinides. That the Wiberg bond index, delocalization index and electron density at the bond critical point show increased bonding in the pyramidal UN3 and US3 structures is consistent with these π effects. Was there really zero participation of f orbitals in the pyramidal and planar structures?

response: All calculations were performed without symmetry constraints, so there are no restrictions on metal-ligand AO mixing in any structure, and so no d orbital is prevented from mixing by symmetry. The lack of symmetry also means that we don't have rigorous sigma and pi MOs, each is a mixture of both, and we have identified them as sigma-type and pi-type by their

principal character (the SI has been modified slightly on page S17 to make this even clearer). It seems logical to have the same bonding pattern, for a given molecule, in both pyr and planar structures, so that we are comparing like for like. The key point is that the changes in NLMO composition in Table S7 between pyr and planar are too small to be structure-directing. Finally, we didn't say there is zero f orbital participation, rather that U uses primarily its 6d orbitals. There is a small contribution from the f-orbitals which agree with the referee's other comment.

Bottom line: great paper.

Mark Schofield

Reviewer #2 (Remarks to the Author):

The solid-state structures of two U(III) complexes, $[U\{N(SiMe_3)_2\}_3]$ and $[U\{S-2,4,6-tBu_3C_6H_2\}_3]$, were examined under high pressure from 0 GPa to 4.09 GPa (for the amide) or 5.26 GPa (for the thiolate). The tris(amide) shows more pyramidalization while the tris(thiolate), surprisingly, shows more planarization under high pressure. Computational methods show this is not due to agostic interactions, as previously shown in the tris(thiolate) complex, but pyramidalization is due to increased dipole and reduction in the volume of the molecule. While planarization in the tris(thiolate) is due to reduction in molecular volume. As the authors points out, planarity is commonly thought to be achieved through more sterically demanding ligands, but this is not the case here with the amide being larger than the thiolate. Overall, I only have some minor changes, and support its publication as it reminds the f element community that things are not just sterically driven.

Minor issues:

1. Caption of Figure 1 - For clarity, I suggest that the authors change to 'out of plane (oop) distance between the M and and the X3 plane, shown in blue,....' I read oop and thought it was a mistake as, at my age, it is usually something I say after sneezing, but that is another matter. I do not believe that oop is defined in the text and probably should be as well.

Corrected.

2. On page 5, SMes* is used as the abbreviation which is correct, just define this earlier for a more general audience who may not know the tris(tert-butyl) phenyl is Mes*.

Now using SAR throughout as we believe it is simpler.

3. The major thing for the authors to chew on is do the authors think that these interactions are agostic or anagostic as defined by Brookhart, Green, and Parkin (PNAS 2007, 104, 6908)? The M-H distances and angles are for transition metals so those distances would naturally be longer with an f element metal center, but the definition of agostic interaction of having some covalent character in a M-H, I would argue, would not apply to any f element compound except with a hydride ligand. Maybe the authors have stronger opinions and it makes no difference if they do not wish to change since it is ingrained in the literature. Again, something for the authors to chew on.

Thank you for these interesting thoughts. One of us had a long Q+A session at a Gordon conference on this subject a few years ago, so it is nice to have the opportunity to give this further thought.

The f-block is also a 'transition series' like the d block, so it is relevant to look for factors that would enable assignment as agostic, rather than anagostic. Even though covalent contributions to bonding remain small in f-block organometallics, as our studies of lower formal oxidation states evolve, we see increasingly behaviors that look less 'electrostatic', and different electron configurations determined by ligand fields, which we used to not think possible outwith the d-block. If in the future we had to switch between agostic and anagostic definitions as we changed oxidation state in an f-block complex, things would become more confusing, and it seems reasonable to retain agostic where possible.

The 'other' 3c2e bonds in which the ligand group really does look more like an H bond donor, with different angles to what we, and d-block chemists observe, still appear to be in the opposite category. For now.

Reviewer #3 (Remarks to the Author):

The authors have presented a thorough and very detailed study regarding U(III) complexes with N- and S-donors. They have used SC-XRD and quantum chemical calculations to analyse the complexes' structural and electronic alterations when applying pressure. They could separate and quantify possible factors influencing a pyramidal vs. planar geometry of the UX₃ compounds, presenting a unique example of a U(III) S-donor complex showing a tendency of planarization upon pressurisation.

The provided manuscript comprises a variety of reliable and very interesting data using non-standard equipment and addressing highly relevant questions in actinide research.

The reviewer had high expectations of the outcome and of the kind of conclusions that might be drawn from the studies, particularly as the introduction also suggested to give answers to urgent questions, e.g. to actinide covalency. However, the expectations could not fully be met. For example, it is noted in the introduction that immense differences in covalency had already been observed when comparing Eu(III) with Am(III). This raises the question why the authors haven't included e.g. Nd(III) complexes NdN₃/ NdS₃ as lanthanide analogues for U(III) in order to get one step closer to resolving this issue (or La(III) for even more comparable ionic radii).

This is an interesting suggestion, which we'd love to pursue in the future, beyond the scope of this word-limited study. We chose to study 5f rather than 4f as this would enable us to see if there was a significant covalency aspect. But as we didn't find this, yes the extension to Nd will be interesting.

Similarly, the reviewer has asked her-/himself why UN₃ was compared with US₃, not N- vs. P-donors or O vs. S.

We have studied the UO₃ system, but it has so many complicated phase transitions and VT as well as VP behaviours caused by changes in the stacking arrangements of the molecules in the lattice, which meant that the symmetrical, well-behaved UN₃ is a much better comparison, and one which we can definitively confirm that is capable of planarization. Additionally, UN₃ is

closely related to the analogue where bulkier groups had been used to explain why that molecule was planar, which teaches us all an important lesson about our preconceptions about how to design molecules.

The P donor analogue would take a lot of additional synthetic work, and does not yet exist for any 4f or 5f metal ion, whereas the MS3 has been reported for both U and a wide range of Ln(III) cations, which we anticipated would make it of more general interest.

Thus, we reviewer would wish to find a slightly revised introduction explaining the necessity of studying geometric changes in An compounds bearing in mind the outcome of this manuscript. In other words, seeking an answer to a clear question at the beginning would help to keep an inner logic of the whole story.

We considered that the sentence on lines 1 and 2 of page 2, that points out that geometry can control electronic structure which can lead to potential applications in quantum sensing and information storage was a pretty strong driver for wanting to understand and control structural changes. However, we are happy to have revised the introduction to lead the reader better through the aims and outcomes of the work.

Still, it is a fascinating and highly relevant work and opens up a broad range of possibilities for future research.

My detailed comments including some typos and minor wording issues to the manuscript are the following:

- 1) In the author's line, a comma is missing after Nikolas Kaltsoyannis. ✓
- 2) Abstract, line 5: There seems to be an unnecessary comma after (UN₃, N" = N(Si(me₃)₂), and after "ambient" there should be "pressure" included. ✓
- 3) Page 2, line 3: ..."to account for this..." "this" should be replaced by e.g. "an" as reference is lacking. ✓
- 4) Introduction, line 14: It should be X(p) pi (with the pi outside the bracket) ✓
- 5) Introduction, line 15, bracket: M = S ... Should it be S_m instead? ✓
- 6) In the introduction, factors for increasing the tendency for pyramidalization are listed. What about strong metal-ligand bonds?

We are not clear what the referee means by this, and are happy to defer to the editor here. To us, it would seem that strong bonding is already included in the various arguments already put forward by the community - short bonds that generate ligand-ligand interactions, and/or multiple, higher-order symmetry bonding interactions.

- 7) Page 3: The single images and the respective text in Figure 1 should be aligned. ✓
- 8) Page 4, Scheme 1: Here, a table with i.a. bond length ranges for UN₃ and US₃ are given. Please double-check the upper value of M-X for US₃ at ambient pressure (2.7143(18)), in comparison with Table S4 (here, it is 3.035(6) Å). Apologies, corrected in the SI.
- 9) Sometimes, there is a space between value and Å, sometimes not. Please unify. ✓
- 10) Figure 3: Something went wrong in the formatting, I guess... Single images should be

sorted and aligned. Apologies. Corrected.

11) Page 6

line 5: tris-thiophenolate instead of trithiophenolate ✓

line 6: It most probably should be “M-S-Cipso” angles? (in the SI, page S-9, it is also displayed like this) ✓

lines 8/9: Reference error for Fig 3 ✓

line 9: Missing space in “, top-left)(average...” ✓

line 9: The pressure symbol is a small p (...at ambient T, P) ✓

line 24: The “t” in tBu should be superscript in the whole text. ✓

All above typos and formatting have been corrected. We thank the reviewer for pointing them out.

line 27 et seq.: It should be mentioned that many calculations were carried out, for ambient but also for high pressure structures.

With respect, we believe we have pointed this out in the text, the methods, and the SI. We would be happy to discuss more with the editor how we can improve this. We would rather not disrupt the flow of the text at line 27 though.

12) It has puzzled me, that US3 was reduced in size for computational studies to US3m. Of course, fewer atoms mean significant reduction of computational costs. On the other hand, US3 is not super big, removal of three electron donating groups will have an electronic impact, and, most importantly, potential agostic interactions would form between the C-H groups of the tBu groups and U, as described in the article. In order to study if agostic interactions play a role or not, tBu groups cannot be removed

Thus it is not surprising that significantly reduced agostic interactions were found in US3m vs UN3 (page 7, line 20 et seq.). In addition, it is no surprise that sulphur containing hydrogen bond interactions also don't play a role in US3m, as they would come from tBu H's, too (page 7, line 25 et seq.).

We think there is a misunderstanding over which tBu groups were removed. It was only the para tBu. This is noted in the manuscript.

13) Page 7, line 19: The “pi” symbol is not displayed.

Apologies, corrected.

14) Page 8, line 12 et seq.: What is the driving force for volume reduction? At least in my understanding it is rather a response to an electronic or magnetic effect. Could you relate the reduced volume e.g. to reduced dipole or electrostatic interactions? Maybe ELF plots can give an answer here in order to visualize electron density changes between ambient and high pressure structures?

The driving force to volume reduction comes from the PV term in $G = U + PV - TS$, ie there is always a drive towards lower volume at high pressure.

15) Page 9, lines 5&6: The introduction suggested an impact of bonding changes and planarization...

Agreed. We hope the revisions to the introduction on the potential of suitably controlled molecules now cover this. In the future, such possibilities may exist for these molecules.

16) Supporting information: A lot of additional data and information is given here, what is really nice and supportive. Please double-check formatting, also including figures and tables. Page S-7, 2nd paragraph, 2nd sentence: "At ambient pressure...". Here, "The structure at ambient pressure" would make more sense.

The following changes have been made to the SI in line with the reviewer's comments:

- Formatting unified with the manuscript to change all references to SMes* to SAR
- All Å signs now have a space preceding them
- The tables (aside from the crystallographic tables) have had their fonts updated to match the main text font.
- Ambient pressure U-S3 distance in table S11 has been updated to correct length from the cif (2.057(19) Å)
- 'Structure at ambient pressure' added page S7 paragraph 2 as requested
- Fig S3 GPa added to axis label of right-hand graph
- Fig S5 GPa added to figure description
- Fig S6 caption: 'at ambient' changed to 'at ambient pressure'
- Fig S8 GPa added to figure description
- Table S4: caption altered from '...for the US3 structures at 5.26 GPa...' to "...for the US3 structures at ambient pressure and at 5.26 GPa...'
- Throughout SI: ortho tert butyl and para tert butyl replaced with ortho-*tert*-butyl and para-*tert*-butyl
- Minor alterations to the descriptions of bonding in the U-N and U-S bonding analysis section on page S-17. The NLMOs are now described as σ -type and π -type. Also a sentence has been added to explain that the use of these terms here." The lack of symmetry precludes rigorous separation into σ and π MOs, and hence we use the designations " σ -type" and " π -type", identifying orbitals by their principal character."
- References style has been unified.

REVIEWERS' COMMENTS

Reviewer #1 (Remarks to the Author):

I was in favor of publication of this manuscript in the first version that I reviewed last month. The minor concerns that I expressed were completely addressed by the authors in their written response as well as in specific changes to the manuscript. I see no barriers to publication.

Reviewer #3 (Remarks to the Author):

There are no more open questions from my side, thanks to the explanations and corrections made by the authors, and I would like to congratulate them again to this nice work. Regarding my concern about the missing tBu groups (point 12). Indeed, there was a misunderstanding from my side. Thus, I take back my criticism and am happy with the reduction of US3 to US3m, which is obviously not changing the outcome of the calculated interactions drastically.

Reviewer response document

Reviewer #1 (Remarks to the Author):

I have reviewed "Contrasting behaviour under pressure reveals the reasons for pyramidalization in tris(amido)uranium(III) and tris(arylthiolate) uranium(III) molecules" by Price and co-workers and am pleased to recommend it for publication. The paper describes an interesting problem - the geometric preferences in actinide complexes bearing multiply bonded ligands - and the authors bring to bear a powerful suite of tools combining synthesis, high pressure crystallography and computational methods to explain the observed structures. Naively, I would have argued that metal-to-ligand π bonding was the critical element of these structures (I would be wrong) and, to my delight, the authors demonstrate convincingly that a combination of polarization, volume minimization and agostic interactions are the key drivers of the observed structures. It's refreshing to see the precision obtainable, for example, of the pyramidal inversion and detailed breakdown that the contributions from agostic interactions make (it was not possible/practical not that long ago).

With these plusses out of the way, here are a few points the authors may wish to consider:

1. Scheme 1 (MS, p.4), shows a reduction in the M-N bond distance and a lengthening of the M-S bond length with increased pyramidalization. Are these both interpreted as a changes of M(d)-X(p) π overlap as suggested in Figure 1(d)? It looks like the M-N π bonds are getting stronger (shorter, higher % contrib. from metal d orbitals) and, regardless, the π system is more complicated than this single figure suggests. True, the in-plane overlap changes with increased pyramidalization, but so does the nature of the d orbitals involved in the bonding. When you depart from D3h, the two sets of doubly-degenerate d-orbitals merge into a set of four, and all can participate in π bonding.

response: The M-N distance does shorten, but by only 0.01 Ang, which is a very small amount. The U-S bond length changes differ – U-S1 lengthens by circa 0.03 Ang., while the other two shorten by less than 0.01 A. Thus it's hard to see how this is explained by π -overlap arguments.

2. On S-17, the authors describe their NBO analysis. I agree with the outcome but the assignment of 1 σ and 1 π bond per amide group is valid only for the pyramidal structure and not the trigonal planar one. In D3h symmetry, the in-plane π orbitals transform as $a_2' + e'$ (see JACS 1990, 112, 1642 for example) and, since it is the d-orbitals that are more important for bonding in early actinides, there are only 2 π bonds that are possible (of e' symmetry). It would not be problematic if f orbitals were used as they span both representations, again affirming the importance of d-orbitals in early actinides. That the Wiberg bond index, delocalization index and electron density at the bond critical point show increased bonding in the pyramidal UN3 and US3 structures is consistent with these π effects. Was there really zero participation of f orbitals in the pyramidal and planar structures?

response: All calculations were performed without symmetry constraints, so there are no restrictions on metal-ligand AO mixing in any structure, and so no d orbital is prevented from mixing by symmetry. The lack of symmetry also means that we don't have rigorous sigma and pi MOs, each is a mixture of both, and we have identified them as sigma-type and pi-type by their

principal character (the SI has been modified slightly on page S17 to make this even clearer). It seems logical to have the same bonding pattern, for a given molecule, in both pyr and planar structures, so that we are comparing like for like. The key point is that the changes in NLMO composition in Table S7 between pyr and planar are too small to be structure-directing. Finally, we didn't say there is zero f orbital participation, rather that U uses primarily its 6d orbitals. There is a small contribution from the f-orbitals which agree with the referee's other comment.

Bottom line: great paper.

Mark Schofield

Reviewer #2 (Remarks to the Author):

The solid-state structures of two U(III) complexes, $[U\{N(SiMe_3)_2\}_3]$ and $[U\{S-2,4,6-tBu_3C_6H_2\}_3]$, were examined under high pressure from 0 GPa to 4.09 GPa (for the amide) or 5.26 GPa (for the thiolate). The tris(amide) shows more pyramidalization while the tris(thiolate), surprisingly, shows more planarization under high pressure. Computational methods show this is not due to agostic interactions, as previously shown in the tris(thiolate) complex, but pyramidalization is due to increased dipole and reduction in the volume of the molecule. While planarization in the tris(thiolate) is due to reduction in molecular volume. As the authors points out, planarity is commonly thought to be achieved through more sterically demanding ligands, but this is not the case here with the amide being larger than the thiolate. Overall, I only have some minor changes, and support its publication as it reminds the f element community that things are not just sterically driven.

Minor issues:

1. Caption of Figure 1 - For clarity, I suggest that the authors change to 'out of plane (oop) distance between the M and and the X3 plane, shown in blue,....' I read oop and thought it was a mistake as, at my age, it is usually something I say after sneezing, but that is another matter. I do not believe that oop is defined in the text and probably should be as well.

Corrected.

2. On page 5, SMes* is used as the abbreviation which is correct, just define this earlier for a more general audience who may not know the tris(tert-butyl) phenyl is Mes*.

Now using SAR throughout as we believe it is simpler.

3. The major thing for the authors to chew on is do the authors think that these interactions are agostic or anagostic as defined by Brookhart, Green, and Parkin (PNAS 2007, 104, 6908)? The M-H distances and angles are for transition metals so those distances would naturally be longer with an f element metal center, but the definition of agostic interaction of having some covalent character in a M-H, I would argue, would not apply to any f element compound except with a hydride ligand. Maybe the authors have stronger opinions and it makes no difference if they do not wish to change since it is ingrained in the literature. Again, something for the authors to chew on.

Thank you for these interesting thoughts. One of us had a long Q+A session at a Gordon conference on this subject a few years ago, so it is nice to have the opportunity to give this further thought.

The f-block is also a 'transition series' like the d block, so it is relevant to look for factors that would enable assignment as agostic, rather than anagostic. Even though covalent contributions to bonding remain small in f-block organometallics, as our studies of lower formal oxidation states evolve, we see increasingly behaviors that look less 'electrostatic', and different electron configurations determined by ligand fields, which we used to not think possible outwith the d-block. If in the future we had to switch between agostic and anagostic definitions as we changed oxidation state in an f-block complex, things would become more confusing, and it seems reasonable to retain agostic where possible.

The 'other' 3c2e bonds in which the ligand group really does look more like an H bond donor, with different angles to what we, and d-block chemists observe, still appear to be in the opposite category. For now.

Reviewer #3 (Remarks to the Author):

The authors have presented a thorough and very detailed study regarding U(III) complexes with N- and S-donors. They have used SC-XRD and quantum chemical calculations to analyse the complexes' structural and electronic alterations when applying pressure. They could separate and quantify possible factors influencing a pyramidal vs. planar geometry of the UX₃ compounds, presenting a unique example of a U(III) S-donor complex showing a tendency of planarization upon pressurisation.

The provided manuscript comprises a variety of reliable and very interesting data using non-standard equipment and addressing highly relevant questions in actinide research. The reviewer had high expectations of the outcome and of the kind of conclusions that might be drawn from the studies, particularly as the introduction also suggested to give answers to urgent questions, e.g. to actinide covalency. However, the expectations could not fully be met. For example, it is noted in the introduction that immense differences in covalency had already been observed when comparing Eu(III) with Am(III). This raises the question why the authors haven't included e.g. Nd(III) complexes NdN₃/ NdS₃ as lanthanide analogues for U(III) in order to get one step closer to resolving this issue (or La(III) for even more comparable ionic radii).

This is an interesting suggestion, which we'd love to pursue in the future, beyond the scope of this word-limited study. We chose to study 5f rather than 4f as this would enable us to see if there was a significant covalency aspect. But as we didn't find this, yes the extension to Nd will be interesting.

Similarly, the reviewer has asked her-/himself why UN₃ was compared with US₃, not N- vs. P-donors or O vs. S.

We have studied the UO₃ system, but it has so many complicated phase transitions and VT as well as VP behaviours caused by changes in the stacking arrangements of the molecules in the lattice, which meant that the symmetrical, well-behaved UN₃ is a much better comparison, and one which we can definitively confirm that is capable of planarization. Additionally, UN₃ is

closely related to the analogue where bulkier groups had been used to explain why that molecule was planar, which teaches us all an important lesson about our preconceptions about how to design molecules.

The P donor analogue would take a lot of additional synthetic work, and does not yet exist for any 4f or 5f metal ion, whereas the MS3 has been reported for both U and a wide range of Ln(III) cations, which we anticipated would make it of more general interest.

Thus, we reviewer would wish to find a slightly revised introduction explaining the necessity of studying geometric changes in An compounds bearing in mind the outcome of this manuscript. In other words, seeking an answer to a clear question at the beginning would help to keep an inner logic of the whole story.

We considered that the sentence on lines 1 and 2 of page 2, that points out that geometry can control electronic structure which can lead to potential applications in quantum sensing and information storage was a pretty strong driver for wanting to understand and control structural changes. However, we are happy to have revised the introduction to lead the reader better through the aims and outcomes of the work.

Still, it is a fascinating and highly relevant work and opens up a broad range of possibilities for future research.

My detailed comments including some typos and minor wording issues to the manuscript are the following:

- 1) In the author's line, a comma is missing after Nikolas Kaltsoyannis. ✓
- 2) Abstract, line 5: There seems to be an unnecessary comma after (UN₃, N" = N(Si(me₃)₂), and after "ambient" there should be "pressure" included. ✓
- 3) Page 2, line 3: ..."to account for this..." "this" should be replaced by e.g. "an" as reference is lacking. ✓
- 4) Introduction, line 14: It should be X(p) pi (with the pi outside the bracket) ✓
- 5) Introduction, line 15, bracket: M = S ... Should it be S_m instead? ✓
- 6) In the introduction, factors for increasing the tendency for pyramidalization are listed. What about strong metal-ligand bonds?

We are not clear what the referee means by this, and are happy to defer to the editor here. To us, it would seem that strong bonding is already included in the various arguments already put forward by the community - short bonds that generate ligand-ligand interactions, and/or multiple, higher-order symmetry bonding interactions.

- 7) Page 3: The single images and the respective text in Figure 1 should be aligned. ✓
- 8) Page 4, Scheme 1: Here, a table with i.a. bond length ranges for UN₃ and US₃ are given. Please double-check the upper value of M-X for US₃ at ambient pressure (2.7143(18)), in comparison with Table S4 (here, it is 3.035(6) Å).Apologies, corrected in the SI.
- 9) Sometimes, there is a space between value and Å, sometimes not. Please unify. ✓
- 10) Figure 3: Something went wrong in the formatting, I guess... Single images should be

sorted and aligned. Apologies. Corrected.

11) Page 6

line 5: tris-thiophenolate instead of trithiophenolate ✓

line 6: It most probably should be “M-S-Cipso” angles? (in the SI, page S-9, it is also displayed like this) ✓

lines 8/9: Reference error for Fig 3 ✓

line 9: Missing space in “, top-left)(average...” ✓

line 9: The pressure symbol is a small p (...at ambient T, P) ✓

line 24: The “t” in tBu should be superscript in the whole text. ✓

All above typos and formatting have been corrected. We thank the reviewer for pointing them out.

line 27 et seq.: It should be mentioned that many calculations were carried out, for ambient but also for high pressure structures.

With respect, we believe we have pointed this out in the text, the methods, and the SI. We would be happy to discuss more with the editor how we can improve this. We would rather not disrupt the flow of the text at line 27 though.

12) It has puzzled me, that US3 was reduced in size for computational studies to US3m. Of course, fewer atoms mean significant reduction of computational costs. On the other hand, US3 is not super big, removal of three electron donating groups will have an electronic impact, and, most importantly, potential agostic interactions would form between the C-H groups of the tBu groups and U, as described in the article. In order to study if agostic interactions play a role or not, tBu groups cannot be removed

Thus it is not surprising that significantly reduced agostic interactions were found in US3m vs UN3 (page 7, line 20 et seq.). In addition, it is no surprise that sulphur containing hydrogen bond interactions also don't play a role in US3m, as they would come from tBu H's, too (page 7, line 25 et seq.).

We think there is a misunderstanding over which tBu groups were removed. It was only the para tBu. This is noted in the manuscript.

13) Page 7, line 19: The “pi” symbol is not displayed.

Apologies, corrected.

14) Page 8, line 12 et seq.: What is the driving force for volume reduction? At least in my understanding it is rather a response to an electronic or magnetic effect. Could you relate the reduced volume e.g. to reduced dipole or electrostatic interactions? Maybe ELF plots can give an answer here in order to visualize electron density changes between ambient and high pressure structures?

The driving force to volume reduction comes from the PV term in $G = U + PV - TS$, ie there is always a drive towards lower volume at high pressure.

15) Page 9, lines 5&6: The introduction suggested an impact of bonding changes and planarization...

Agreed. We hope the revisions to the introduction on the potential of suitably controlled molecules now cover this. In the future, such possibilities may exist for these molecules.

16) Supporting information: A lot of additional data and information is given here, what is really nice and supportive. Please double-check formatting, also including figures and tables. Page S-7, 2nd paragraph, 2nd sentence: "At ambient pressure...". Here, "The structure at ambient pressure" would make more sense.

The following changes have been made to the SI in line with the reviewer's comments:

- Formatting unified with the manuscript to change all references to SMes* to SAR
- All Å signs now have a space preceding them
- The tables (aside from the crystallographic tables) have had their fonts updated to match the main text font.
- Ambient pressure U-S3 distance in table S11 has been updated to correct length from the cif (2.057(19) Å)
- 'Structure at ambient pressure' added page S7 paragraph 2 as requested
- Fig S3 GPa added to axis label of right-hand graph
- Fig S5 GPa added to figure description
- Fig S6 caption: 'at ambient' changed to 'at ambient pressure'
- Fig S8 GPa added to figure description
- Table S4: caption altered from '...for the US3 structures at 5.26 GPa...' to "...for the US3 structures at ambient pressure and at 5.26 GPa...'
- Throughout SI: ortho tert butyl and para tert butyl replaced with ortho-*tert*-butyl and para-*tert*-butyl
- Minor alterations to the descriptions of bonding in the U-N and U-S bonding analysis section on page S-17. The NLMOs are now described as σ -type and π -type. Also a sentence has been added to explain that the use of these terms here." The lack of symmetry precludes rigorous separation into σ and π MOs, and hence we use the designations " σ -type" and " π -type", identifying orbitals by their principal character."
- References style has been unified.